# DEGREE: Decomposition Based Explanation for Graph Neural Networks

**Qizhang Feng[1], Ninghao Liu[2], Fan Yang[3], Ruixiang Tang[3], Mengnan Du[1], Xia Hu[3]**
[1]Department of Computer Science and Engineering, Texas A&M University
[2]Department of Computer Science, University of Georgia
[3]Department of Computer Science, Rice University
{qf31,dumengnan}@tamu.edu, ninghao.liu@uga.edu, {fy19,rt39,xia.hu}@rice.edu

## ABSTRACT

Graph Neural Networks (GNNs) are gaining extensive attention for their application in graph data. However, the black-box nature of GNNs prevents users from understanding and trusting the models, thus hampering their applicability. Whereas explaining GNNs remains a challenge, most existing methods fall into approximation based and perturbation based approaches with suffer from faithfulness problems and unnatural artifacts, respectively. To tackle these problems, we propose DEGREE (Decomposition based Explanation for GRaph nEural nEtworks) to provide a faithful explanation for GNN predictions. By decomposing the information generation and aggregation mechanism of GNNs, DEGREE allows tracking the contributions of specific components of the input graph to the final prediction. Based on this, we further design a subgraph level interpretation algorithm to reveal complex interactions between graph nodes that are overlooked by previous methods. The efficiency of our algorithm can be further improved by utilizing GNN characteristics. Finally, we conduct quantitative and qualitative experiments on synthetic and real-world datasets to demonstrate the effectiveness of DEGREE on node classification and graph classification tasks.

## 1 INTRODUCTION

Graph Neural Networks (GNNs) play an important role in modeling data with complex relational information (Zhou et al., 2018), which is crucial in applications such as social networking (Fan et al., 2019), advertising recommendation (Liu et al., 2019), drug generation (Liu et al., 2020), and agent interaction (Casas et al., 2019). However, GNN suffers from its black-box nature and lacks a faithful explanation of its predictions.

Recently, several approaches have been proposed to explain GNNs. Some of them leverage gradient or surrogate models to approximate the local model around the target instance (Huang et al., 2020; Baldassarre & Azizpour, 2019; Pope et al., 2019). Some other methods borrow the idea from perturbation based explanation (Ying et al., 2019; Luo et al., 2020; Lucic et al., 2021), under the assumption that removing the vital information from input would significantly reduce output confidence. However, approximation based methods do not guarantee the fidelity of the explanation obtained, as Rudin (2019) states that a surrogate that mimics the original model possibly employs distinct features. On the other hand, perturbation based approaches may trigger the adversarial nature of deep models. Chang et al. (2018) reported this phenomenon where masking some parts of the input image introduces unnatural artifacts. Additionally, additive feature attribution methods (Vu & Thai, 2020; Lundberg & Lee, 2017) such as gradient based methods and GNNExplainer only provide a single heatmap or subgraph as explanation. The nodes in graph are usually semantically individual and we need a fine-grained explanation to the relationships between them. For example, in organic chemistry, the same functional group combined with different structures can exhibit very different properties.

To solve the above problems, we propose DEGREE (Decomposition based Explanation for GRaph nEural nEtworks) , which measures the contribution of components in the input graph to the GNN prediction. Specifically, we first summarize the intuition behind the Context Decomposition (CD)

algorithm (Murdoch et al., 2018) and propose that the information flow in GNN's message propagation mechanism is decomposable. Then we design the decomposition schemes for the most commonly used layers and operations in GNNs, so as to isolate the information flow from distinct node groups. Furthermore, we explore the subgraph-level explanation via an aggregation algorithm that utilizes DEGREE and the structural information of the input graph to construct a series of subgraph sets as the explanation. DEGREE guarantees explanation fidelity by directly analyzing GNN feed-forward propagation, instead of relying on input perturbation or the use of alternative models. DEGREE is non-additive and can therefore uncover non-linear relationships between nodes. We quantitatively and qualitatively evaluate the DEGREE on both synthetic and real-world datasets to validate the effectiveness of our method. The contributions of this work are summarized as follows:

- We propose a new explanation method (DEGREE) for GNNs, from the perspective of decomposition. By elucidating the feed-forward propagation mechanism within GNN, DEGREE allows capturing the contribution of individual components of the input graph to the final prediction.
- We propose an aggregation algorithm that provides important subgraphs as explanation in order to mine the complex interactions between graph components. We combine the property of the message propagation mechanism to further reduce the computation.
- We evaluate DEGREE on both synthetic and real-world datasets. The quantitative experiments show that our method could provide faithful explanations. The qualitative experiments indicate that our method may capture the interaction between graph components.

## 2 RELATED WORK

Despite the great success in various applications, the black-box nature of deep models has long been criticized. Explainable Artificial Intelligence (XAI) tries to bridge the gap by understanding the internal mechanism of deep models (Du et al., 2019). Meanwhile, the need to tackle non-Euclidean data, such as geometric information, social networks, has given rise to the development of GNNs. Similar to the tasks on image or text data, GNNs focus on node classification (Henaff et al., 2015), graph classification (Xu et al., 2018; Zhang et al., 2018), and link prediction (Zhang & Chen, 2018; Cai & Ji, 2020). Message passing mechanism allows the information to flow from one node to another along edges, and empowers GNNs with convolutional capabilities for graph data.

While the explainability in image and text domains is widely studied (Shrikumar et al., 2017; Sundararajan et al., 2017; Simonyan et al., 2013), the explainability of GNN is on the rise. First, some recent work adapts the interpretation methods used for traditional CNNs to GNNs (Baldassarre & Azizpour, 2019; Pope et al., 2019). They employ gradient values to investigate the contribution of node features to the final prediction. However, these methods ignore the topological information, which is a crucial property of graph data. Second, some methods trace the model prediction back to the input space in a backpropagation manner layer by layer (Schwarzenberg et al., 2019; Schnake et al., 2020). Third, some methods define a perturbation-based interpretation whereby they perturb node, edge, or node features and identify the components that affect the prediction most. Specifically, GNNExplainer and PGExplainer (Ying et al., 2019) maximize the mutual information between perturbed input and original input graph to identify the important features. Causal Screening (Wang et al., 2021) searches for the important subgraph by monitoring the mutual information from a cause-effect standpoint. CF-GNNExplainer (Lucic et al., 2021) proposes to generate counterfactual explanations by finding the minimal number of edges to be removed such that the prediction changes. In addition, XGNN (Yuan et al., 2020) builds a model-level explanation for GNNs by generating a prototype graph that can maximize the prediction. Moreover, due to the discrete and topological nature of graph data, XGNN defines graph generation as a reinforcement learning task instead of gradient ascent optimization.

Many previous explanation methods for GNNs suffer from adversarial triggering issues, faithfulness issues and additive assumptions. To this end, we propose a decomposition based explanation for GNNs (DEGREE) to remedy these problems. DEGREE enables to track the contribution of the components from the input graph to the final prediction by decomposing a trained GNN. Thus, DEGREE guarantees the integrity of the input and eliminates the adversarial triggering issue of the perturbation-based approach. Since no surrogate models are used, DEGREE guarantees its faithfulness. Meanwhile, by integrating the decomposition to the normal layer, DEGREE does not have any additional training process.

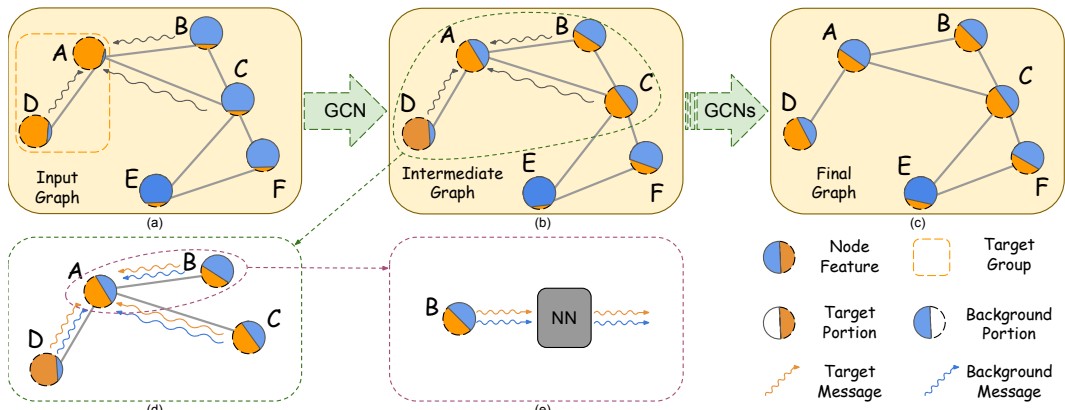

Figure 1: Illustration of the DEGREE for decomposing GCN. Node features or latent embeddings contain *target* portion (orange hemisphere) and an *background* portion (blue hemisphere). (a)-(c) show the workflow of the GCN, exhibiting only the messages aggregation for node A. (d) demonstrates message aggregation after decomposition. (e) demonstrates the decomposed message flow.

## 3 DEGREE: DECOMPOSITION BASED EXPLANATION FOR GNNS

In this section, we introduce the details of the proposed explanation method. First, we introduce the notations and problem definition. Then, we discuss the general idea of decomposition based explanation. Finally, we develop concrete decomposition schemes for different GNNs layers.

### 3.1 PROBLEM DEFINITION

We first introduce the notations used in this work. Given a graph $\mathcal{G} = (\mathcal{V}, \mathcal{E})$, where $\mathcal{V}$ is the set of nodes and $\mathcal{E}$ is the set of edges between nodes. The adjacency matrix of $\mathcal{G}$ is denoted as $\mathbf{A} \in \mathbb{R}^{N \times N}$, where $N = |\mathcal{V}|$ is the number of nodes, so $\mathcal{V} = \{v_1, v_2, ..., v_N\}$. The nodes are associated with features, and the feature matrix is denoted as $\mathbf{X} \in \mathbb{R}^{N \times F}$, where $F$ is the feature dimension.

In this work, we focus on explaining GNN-based classification models. Let $f$ denote the target GNN model. $f$ computes the likelihood of a node or graph belonging to the target class, where $f : \mathcal{G} \mapsto \mathbb{R}^{|\mathcal{V}|}$ or $f : \mathcal{G} \mapsto \mathbb{R}$, for node or graph classification respectively.

The *goal* of explanation is to find the most important subgraph in $\mathcal{G}$ given $f(\mathcal{G})$, which requires measuring the contribution of all possible subgraphs and find the ones with high contribution scores. However, there are two challenges to be addressed. (1) Given any subgraph of interest, how to estimate its contribution without breaking up the input graph? (2) The number of all subgraphs in $\mathcal{G}$ is usually very large, so how to choose candidate subgraphs for improving explanation efficiency? We tackle the first challenge in the following part of Sec 3 and solve the second one in Sec 4.

### 3.2 DECOMPOSITION BASED EXPLANATION

In general, a prediction model $f$ contains multiple layers $L_t, t \in \{1, \ldots, T\}$:

$$f(\mathbf{X}) = L_T \circ L_{T-1} \circ \cdots \circ L_2 \circ L_1(\mathbf{X}). \tag{1}$$

Let $\mathbf{X}[t]$ denotes the input to $L_t$, so $\mathbf{X}[t+1] = L_t(\mathbf{X}[t])$ and $\mathbf{X}[1] = \mathbf{X}$. The symbol $\circ$ denotes function composition. Here $\mathbf{X}[t] \in \mathbb{R}^{N \times F_t}$ is the embedding matrix at $t$-th layer, where $F_t$ is the latent dimension. The embedding vector of the $i$-th node at $t$-th layer is denoted as $\mathbf{X}_i[t]$.

The core idea of decomposition based explanation is that, given a target node group (or subgraph) of interest, we estimate its contribution score to model prediction merely through feed-forward propagation. We call the information propagated from the target group as *target* portion, and the rest of information is called *background* portion. It is worth noting that, a node is in the target group does not necessarily mean it is important, while it only means we are interested in its importance score. In the following, we use $\gamma$ and $\beta$ to denote the target and background portion, respectively. Let $\mathbf{m} \in \{0, 1\}^N$, where $\mathbf{m}_i = 1$ means $v_i$ belongs to the target group and otherwise $\mathbf{m}_i = 0$.

Then, the decomposition is initialized from the layer of node features, where the target portion and background portion of the input feature matrix are: $\mathbf{X}^\gamma = diag(\mathbf{m})\mathbf{X}$ and $\mathbf{X}^\beta = (\mathbf{I} - diag(\mathbf{m}))\mathbf{X}$, respectively. In a neural network, information from different parts of input are merged in the feed-forward process into latent representations, which poses challenges for explanation. Suppose the target and background portion in $\mathbf{X}[t]$ are known from prior layer, we could explain the model if we can still distinguish the information flows of the two portions inside $L_t$. That is, at layer $L_t$, suppose its input can be decomposed as $\mathbf{X}[t] = \mathbf{X}^\gamma[t] + \mathbf{X}^\beta[t]$, the following relations need to hold for explanation:

$$L_t^D(\mathbf{X}^\gamma[t], \mathbf{X}^\beta[t]) = \left( \underbrace{\Gamma(\mathbf{X}^\gamma[t], \mathbf{X}^\beta[t])}_{\mathbf{X}^\gamma[t+1]}, \underbrace{B(\mathbf{X}^\gamma[t], \mathbf{X}^\beta[t])}_{\mathbf{X}^\beta[t+1]} \right) \tag{2}$$

$$L_t(\mathbf{X}[t]) = \mathbf{X}[t+1] = \mathbf{X}^\gamma[t+1] + \mathbf{X}^\beta[t+1], \tag{3}$$

where $L_t^D(\cdot, \cdot)$ denotes the decomposed version of layer $L_t$. $\Gamma(\cdot, \cdot)$ and $B(\cdot, \cdot)$ corresponds to the contribution of the target and the background portion to layer $L_t$. $\mathbf{X}^\gamma[t+1]$ and $\mathbf{X}^\beta[t+1]$ denotes the target and background portion of $\mathbf{X}[t+1]$ as the input to the next layer. The decomposition above goes from the input, through all intermediate layers, to the final prediction. If a target node group or subgraph is important, then it should contributes to most of the prediction, meaning that $\Gamma(\mathbf{X}^\gamma[T], \mathbf{X}^\beta[T]) \approx f(\mathbf{X})$.

### 3.3 INTUITIONS BEHIND DECOMPOSITION BASED EXPLANATION FOR GNN

The intuition behind decomposition based explanation could be summarized as two rules: (1) the target and background portion at a higher layer mainly comes from the target and background portion at the lower layer respectively; (2) ideally there should be little interaction between the target portion and the background portion. Please note that the partition is not dimension-wise, meaning that each latent dimension may contain information from both target and background portions.

Figure 1 briefly illustrates the working principle of GNNs: the model computes neural message for each node pair and aggregates message for them from their neighbors. A major step of decomposing GNNs is that: the target and background portion of a node are aggregated from the target and background portion of its neighbours, respectively. This can be easily illustrated by the distributive nature of the GNN information aggregation mechanism:

$$\mathbf{X}[t+1] = \mathbf{A}\mathbf{X}[t] = \mathbf{A}\left(\mathbf{X}^\gamma[t] + \mathbf{X}^\beta[t]\right) = \underbrace{\mathbf{A}\mathbf{X}^\gamma[t]}_{\mathbf{X}^\gamma[t+1]} + \underbrace{\mathbf{A}\mathbf{X}^\beta[t]}_{\mathbf{X}^\beta[t+1]}. \tag{4}$$

Nevertheless, the above equation is only a conceptual illustration. A real GNN model could consist of various layers, such as graph convolution layers, fully connected layers, activation layers and pooling layers. Several challenges still need to be tackled to develop an effective explanation method. First, how to design the decomposition scheme for different types of layers? Second, how to efficiently find out the important nodes and subgraphs, by choosing the appropriate target/background group given all possible node combinations?

### 3.4 DECOMPOSING GNN LAYERS

In this work, we consider the decomposition scheme for two commonly used GNN architectures: GCN (Kipf & Welling, 2016) and GAT (Veličković et al., 2017).

#### 3.4.1 DECOMPOSING GCNS

The GCN architecture consists of graph convolution, fully connected layers, ReLU and maxpooling.

**Graph Convolution Layer:** The graph convolution operation pass messages between nodes:

$$\mathbf{X}[t+1] = \tilde{\mathbf{D}}^{-\frac{1}{2}}\tilde{\mathbf{A}}\tilde{\mathbf{D}}^{-\frac{1}{2}}\mathbf{X}[t]\mathbf{W} + \mathbf{b}, \tag{5}$$

where $\mathbf{W}$ and $\mathbf{b}$ denote the trainable weights and bias. Here $\mathbf{b}$ is optional. $\tilde{\mathbf{A}} = \mathbf{A} + \mathbf{I}$ denotes the adjacency matrix with self loop. The matrix $\tilde{\mathbf{D}}_{i,i} = \sum_j \tilde{\mathbf{A}}_{i,j}$ is the diagonal degree matrix of $\tilde{\mathbf{A}}$. The corresponding decomposition can be designed as follows:

$$\boldsymbol{\gamma}[t] = \tilde{\mathbf{D}}^{-\frac{1}{2}} \tilde{\mathbf{A}} \tilde{\mathbf{D}}^{-\frac{1}{2}} \mathbf{X}^{\gamma}[t] \mathbf{W} \ , \ \boldsymbol{\beta}[t] = \tilde{\mathbf{D}}^{-\frac{1}{2}} \tilde{\mathbf{A}} \tilde{\mathbf{D}}^{-\frac{1}{2}} \mathbf{X}^{\beta}[t] \mathbf{W}, \tag{6}$$

$$\mathbf{X}^{\gamma}[t+1] = \boldsymbol{\gamma}[t] + \mathbf{b} \cdot \frac{\left| \boldsymbol{\gamma}[t] \right|}{\left| \boldsymbol{\gamma}[t] \right| + \left| \beta[t] \right|} \ , \ \mathbf{X}^{\beta}[t+1] = \boldsymbol{\beta}[t] + \mathbf{b} \cdot \frac{\left| \boldsymbol{\beta}[t] \right|}{\left| \boldsymbol{\gamma}[t] \right| + \left| \beta[t] \right|}, \tag{7}$$

where $\mathbf{X}^{\gamma}[t]$ and $\mathbf{X}^{\beta}[t]$ is the target and background portion of $\mathbf{X}[t]$, respectively. The derivation of $\boldsymbol{\gamma}[t]$ and $\boldsymbol{\beta}[t]$ is intuitive since graph convolution is a linear operation. Motivated by (Singh et al., 2018), $\boldsymbol{\gamma}[t]$ and $\boldsymbol{\beta}[t]$ have to compete for their share of $\mathbf{b}$ as in Eq 7. $\left| \boldsymbol{\gamma}[t] \right| \in \mathbb{R}^{F_{t+1}}$ measures the dimension-wise magnitude of $\mathbf{X}^{\gamma}[t]$ after the linear mapping ($\left| \boldsymbol{\beta}[t] \right|$ is defined similarly).

**Fully Connected Layer:** A fully connected layer prevalent in the model is shown below:

$$\mathbf{X}[t+1] = \mathbf{X}[t]\Theta + \mathbf{b}, \tag{8}$$

where $\Theta$ and $\mathbf{b}$ denote trainable weights and bias. Structure-wise, it is very similar to the GCN. The decomposition can be designed as:

$$\mathbf{X}^{\gamma}[t+1] = \mathbf{X}^{\gamma}[t]\Theta + \mathbf{b} \cdot \frac{\left| \mathbf{X}^{\gamma}[t]\Theta \right|}{\left| \mathbf{X}^{\gamma}[t]\Theta \right| + \left| \mathbf{X}^{\beta}[t]\Theta \right|}, \mathbf{X}^{\beta}[t+1] = \mathbf{X}^{\beta}[t]\Theta + \mathbf{b} \cdot \frac{\left| \mathbf{X}^{\beta}[t]\Theta \right|}{\left| \mathbf{X}^{\gamma}[t]\Theta \right| + \left| \mathbf{X}^{\beta}[t]\Theta \right|}. \tag{9}$$

**ReLU Activation:** For the activation operator ReLU, we use the telescoping sum decomposition from Murdoch & Szlam (2017). We update the target term first and then update the background term by subtracting this from total activation:

$$\mathbf{X}^{\gamma}[t+1] = \text{ReLU}\left( \mathbf{X}^{\gamma}[t] \right), \ \mathbf{X}^{\beta}[t+1] = \text{ReLU}\left( \mathbf{X}^{\gamma}[t] + \mathbf{X}^{\beta}[t] \right) - \text{ReLU}\left( \mathbf{X}^{\gamma}[t] \right). \tag{10}$$

**Maxpooling:** We track the node indices selected by pooling in both target and background portion.

### 3.4.2 DECOMPOSING GATs

The graph attention layer in GAT is similar to Eq. 5, but uses the attention coefficients $\alpha_{i,j}$ to aggregate the information (an alternative way to understand Eq. 5 is that $\alpha_{i,j} = (\tilde{\mathbf{D}}^{-\frac{1}{2}} \tilde{\mathbf{A}} \tilde{\mathbf{D}}^{-\frac{1}{2}})_{i,j}$):

$$\alpha_{i,j} = \frac{\exp\left( \text{LeakyReLU}\left( \left[ \mathbf{X}_i[t]\mathbf{W} \| \mathbf{X}_j[t]\mathbf{W} \right] \mathbf{a} \right) \right)}{\sum_{k \in \mathcal{N}_i \cup \{i\}} \exp\left( \text{LeakyReLU}\left( \left[ \mathbf{X}_i[t]\mathbf{W} \| \mathbf{X}_k[t]\mathbf{W} \right] \mathbf{a} \right) \right)}, \tag{11}$$

where $\|$ represents the concatenation operation. $\mathbf{W}$ and $a$ are parameters. $\mathbf{X}_i[t]$ denotes the embedding of node $i$ at layer $L_t$. $\mathcal{N}_i$ denotes the neighbors of node $i$.

Therefore, a graph attention layer can be seen as consisting of four smaller layers: linear mapping, concatenation, LeakyReLU activation, and softmax operation. Decomposing a linear mapping is as trivial as decomposing an FC layer. To decompose the concatenation operator:

$$\mathbf{X}_i[t] \| \mathbf{X}_j[t] = \mathbf{X}_i^{\gamma}[t] \| \mathbf{X}_j^{\gamma}[t] + \mathbf{X}_i^{\beta}[t] \| \mathbf{X}_j^{\beta}[t]. \tag{12}$$

For LeakyReLU, the idea of decomposition is the same as ReLU. For softmax operation, we split the coefficients proportionally to the exponential value of the target and the background term of input:

$$\mathbf{X}^{\gamma}[t+1] = softmax\left( \mathbf{X}[t] \right) \cdot \frac{\exp\left( \left| \mathbf{X}^{\gamma}[t] \right| \right)}{\exp\left( \left| \mathbf{X}^{\gamma}[t] \right| \right) + \exp\left( \left| \mathbf{X}^{\beta}[t] \right| \right)},$$

$$\mathbf{X}^{\beta}[t+1] = softmax\left( \mathbf{X}[t] \right) \cdot \frac{\exp\left( \left| \mathbf{X}^{\beta}[t] \right| \right)}{\exp\left( \left| \mathbf{X}^{\beta}[t] \right| \right) + \exp\left( \left| \mathbf{X}^{\gamma}[t] \right| \right)}. \tag{13}$$

Here we employ the similar motivation that used to split bias term in Eq. 7, and let $\left| \mathbf{X}^{\gamma}[t] \right|$ and $\left| \mathbf{X}^{\beta}[t] \right|$ to compete for the original value. The detail of decomposing the attention coefficients can be found in Appendix B.

# 4 SUBGRAPH-LEVEL EXPLANATION VIA AGGLOMERATION

Through decomposition, we could compute the contribution score of any given node groups. However, this is not enough for explaining GNNs. Our goal of explanation is to find the most important subgraph structure, but it is usually impossible to exhaustively compute and compare the scores of all possible subgraphs. In this section, we design an agglomeration algorithm to tackle the challenge.

## 4.1 CONTEXTUAL CONTRIBUTION SCORE

We first introduce a new scoring function to be used in our algorithm. Different from the *absolute contribution* scores provided by decomposition, in many scenarios, we are more interested in the *relative contribution* of the target compared to its contexts. Let $\mathcal{V}^\gamma \subset \mathcal{V}$ be the target node group, and $f^D(\cdot)$ be the contribution score calculated from decomposition. The relative contribution of $\mathcal{V}^\gamma$ averaged over different contexts is calculated as:

$$\phi(\mathcal{V}^\gamma) \triangleq \mathbb{E}_{\mathcal{C} \sim RW(\mathcal{N}_L(\mathcal{V}^\gamma))} \left[ f^D(\mathcal{V}^\gamma \cup \mathcal{C}) - f^D(\mathcal{C}) \right], \tag{14}$$

where $\mathcal{C}$ is the context around $\mathcal{V}^\gamma$, and $\mathcal{N}_L(\mathcal{V}^\gamma)$ contains the neighboring nodes of $\mathcal{V}^\gamma$ within $L$-hops. Here we use a random walk process $RW()$ to sample $\mathcal{C}$ within the neighborhood around $\mathcal{V}^\gamma$. The reason for sampling within the neighborhood is based on the information aggregation, where a node collects the information from its neighbors within certain hops constrained by the GNN depth.

## 4.2 SUBGRAPHS CONSTRUCTION VIA AGGLOMERATION

Our agglomeration algorithm initializes from individual nodes and terminates when the whole graph structure is included. Specifically, the interpretation process constructs a series of intermediate subgraph sets $\mathcal{E} = \{\mathcal{S}_1, ..., \mathcal{S}_I\}$, where $\mathcal{S}_i = \{\mathcal{B}_1, ..., \mathcal{B}_{M_i}\}$ contains $M_i$ subgraphs. At each step, the algorithm searches for the candidate node or node group $v$ that most significantly affects the contribution of subgraph $\mathcal{B}_m, m \in \{1, ..., M_i\}$ according to the ranking score $r(v)$:

$$s(v) \triangleq \phi\left(\{v\} \cup \mathcal{B}_m\right) - \phi\left(\mathcal{B}_m\right), r(v) \triangleq \left| s(v) - \mathbb{E}_{v'}\left[s(v')\right] \right|, \ s.t. \ v, v' \in \mathcal{N}(\mathcal{B}_m), \tag{15}$$

where $\mathcal{N}(\mathcal{B}_m)$ is the set of neighbor nodes or node groups to $\mathcal{B}_m$. Here $s(v)$ measures the influence of $v$ to $\mathcal{B}_m$, while $r(v)$ further revises the value by considering the relative influence of $v$ compared to other candidates $v'$. At the beginning of our algorithm, $\mathcal{B}_m = \emptyset$. A node $v$ is selected if $r(v) \geq q \cdot \max_{v'} r(v')$, and we set $q = 0.6$ in experiments. The selected nodes are merged into subgraphs to form $\mathcal{S}_{i+1}$. Small subgraphs will be merged into larger ones, so we have $M_i \leq M_j, i \geq j$. The algorithm executes the above steps repeatedly and terminates until all nodes are included (i.e., $M_i = 1$), or a certain pre-defined step budget is used up. Further details of the algorithm can be found in Section C of the Appendix.

# 5 EXPERIMENTS

## 5.1 EXPERIMENTAL DATASETS

Following the setting in previous work (Ying et al., 2019), we adopt both synthetic datasets and real-world datasets. The statistic of all datasets are given in Sec A in the Appendix.

### 5.1.1 SYNTHETIC DATASETS

- **BA-Shapes.** BA-Shapes is a unitary graph based on a 300-node Barabási-Albert (BA) graph (Barabási & Albert, 1999). 80 five-node motifs are randomly attached to the base graph. The motif is a "house" structured network in which the points are divided into top-nodes, middle-nodes, or bottom-nodes. 10% of random edges are attached to perturb the graph.
- **BA-Community.** BA-Community dataset is constructed by combining two BA-Shapes graphs. To distinguish the nodes, the distribution of features of nodes in different communities differs. There are eight node classes based on the structural roles and community membership.

- **Tree-Cycles.** The Tree-Cycles dataset germinates from an eight-level balanced binary tree base graph. The motif is a six-node cycle. 80 motifs are randomly added to the nodes of the base graph. The nodes are classified into two classes, i.e., base-nodes and motif-nodes.
- **Tree-Grids.** It is constructed in the same way as the Tree-Cycles dataset. The Tree-Grid dataset has the same base graph while replacing the cycle motif with a 3-by-3 grid motif.

### 5.1.2 REAL-WORLD DATASETS

- **MUTAG.** It is a dataset with 4,337 molecule graphs. Every graph is labeled according to their mutagenic effect on the bacterium. As discussed in (Debnath et al., 1991), the molecule with chemical group $NH_2$ or $NO_2$ and carbon rings are known to be mutagenic. Since non-mutagenic molecules have no explicit motifs, only mutagenic ones are presented during the analysis.
- **Graph-SST2.** It is a dataset of 70,042 sentiment graphs, which are converted through Biaffine parser (Liu et al., 2021). Every graph is labeled according to its sentiment, either positive or negative. The nodes denote words, and edges denote their relationships. The node features are initialized as the pre-trained BERT word embeddings (Devlin et al., 2019).

## 5.2 EXPERIMENTAL SETUP

### 5.2.1 EVALUATION METRICS

The interpretation problem is formalized as a binary classification problem distinguishing between important and unimportant structures (nodes or edges, depending on the nature of ground truth). A good explanation should assign high scores to the important structures and low scores to unimportant ones. We consider the nodes within the motif to be important for the synthetic dataset and the rest to be unimportant. In the MUTAG dataset, the "N-H" and "N-O" edges are important, and the rest are unimportant. We conduct quantitative experiments on the synthetic datasets and the MUTAG dataset, and qualitative experiments on the MUTAG dataset and the Graph-SST2 dataset. We adopt the Area Under Curve (AUC) to evaluate the performance quantitatively.

### 5.2.2 BASELINES METHODS AND IMPLEMENTATION DETAILS

**Baselines Methods.** We compare with four baselines methods: GRAD (Ying et al., 2019), GAT (Veličković et al., 2017), GNNExplainer (Ying et al., 2019) and PGExplainer (Luo et al., 2020). (1) GRAD computes the gradients of GNN output with respect to the adjacency matrix or node features. (2) GAT averages the attention coefficients across all graph attention layers as edge importance. (3) GNNExplainer optimizes a soft mask of edges or node features by maximizing the mutual information. (4) PGExplainer learns an MLP (Multi-layer Perceptron) model to generate the mask using the reparameterization trick (Jang et al., 2017).

**Construction of Target Models.** We use all synthetic datasets together with the MUTAG dataset for quantitative evaluation experiments. We train a GCN and GAT model as the model to be explained for all datasets following the setup of previous work. Meanwhile, we construct DEGREE(GCN) and DEGREE(GAT) as the decomposed version for our method. We set the number of GNN layers to 3 for all datasets, except for the Tree-Grid dataset where it is 4. Since the 3-hop neighbors of some target nodes has only in-motif nodes (no negative samples). For the qualitative evaluation experiment, we use the MUTAG dataset and Graph-SST2 dataset. For all the model training, we use Adam optimizer. All the datasets are divided into train/validation/test sets.

**Explainer Setting.** For all baseline methods, we keep the default hyper-parameters. For baselines (e.g., PGExplainer) who need training additional modules, we also split the data. We also split data for baselines requiring additional training (e.g. PGExplainer). We use all nodes in the motif for evaluation. For explainers that only provide node explanation, we average the score of its vertexes as edge explanation. The details of explainer settings can be found in Sec A in Appendix.

## 5.3 QUANTITATIVE EVALUATION

In this section, we introduce experimental results on both synthetic datasets and the MUTAG dataset. For node classification, the computation graph only contains nodes within $l$-hop from the target node, where $l$ is the number of model layer. The reason is that the nodes outside the computation

Table 1: Quantitative Experiment Result.

| | Explanation AUC | | | | |
|---|---|---|---|---|---|
| Task | Node Classification | | | | Graph Classification |
| Dataset | BA-Shapes | BA-Community | Tree-Cycles | Tree-Grid | MUTAG |
| GRAD | 0.882 | 0.750 | 0.905 | 0.612 | 0.787 |
| GAT | 0.815 | 0.739 | 0.824 | 0.667 | 0.763 |
| GNNExplainer | 0.832 | 0.750 | 0.862 | 0.842 | 0.742 |
| PGExplainer | 0.963 | 0.894 | **0.960** | 0.907 | 0.836 |
| DEGREE(GCN) | **0.991**±0.005 | **0.984**±0.005 | 0.958±0.004 | **0.925**±0.040 | **0.875**±0.028 |
| DEGREE(GAT) | 0.990±0.008 | 0.982±0.010 | 0.919±0.027 | 0.935±0.031 | 0.863±0.042 |

| | Time Efficiency(s) | | | | |
|---|---|---|---|---|---|
| GNNExplainer | 0.65 | 0.78 | 0.69 | 0.72 | 0.43 |
| PGExplainer | 116.72(0.014) | 35.71(0.024) | 117.96(0.09) | 251.37(0.011) | 503.52(0.012) |
| DEGREE(GCN) | 0.44 | 1.02 | 0.25 | 0.37 | 0.83 |
| DEGREE(GAT) | 1.98 | 2.44 | 0.96 | 1.03 | 0.79 |

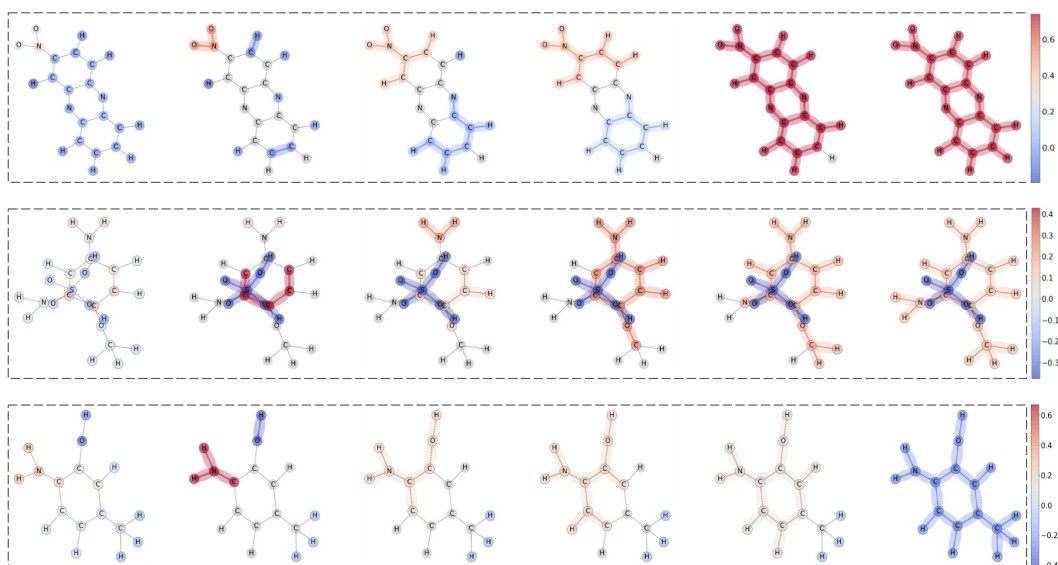

Figure 2: The subgraph agglomeration results on MUTAG dataset. The first row shows a correct prediction. The second and the third row report two typical examples of errors. Red is mutagenic, blue is non-mutagenic, gray is not selected. The colored edges link the selected nodes. The process goes from left to right. The graph on the far left in each row displays the score for individual nodes.

graph will not influence the final prediction. Table 1 shows the explanation AUC and time efficiency (the training time is shown outside the parentheses for PGExplainer). We have the following key findings. First, DEGREE achieves SOTA performances in most scenarios, showing its advantages in faithfulness over baseline methods. Second, the performance of DEGREE on GCN and GAT models can achieve similar high performance. This observation demonstrates the adaptability of our approach. Third, the improvement of AUC on BA-Community (~9%) and MUTAG (~5%) is more noticeable, where the two datasets distinguish themselves from others is that their features are not constants. It thus shows that our explanation method could well handle node features as they propagate through the graph structure. In terms of efficiency, DEGREE is implemented by decomposing the built-in forward propagation function, so there is no training process. The time cost is highly correlated to the complexity of the target model and the input size. We report further quantitative experiments in Appendix D.

## 5.4 QUALITATIVE EVALUATION

In this section, we use Graph-SST2 and MUTAG datasets to visualize the explanation and demonstrate the effectiveness of our subgraph agglomeration algorithm in Sec 4.

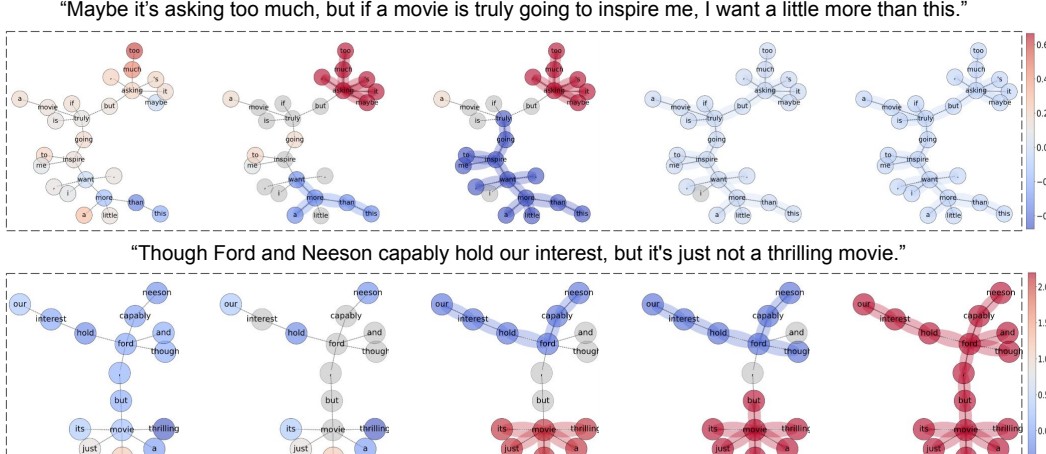

Figure 3: The subgraph agglomeration results on the Graph-SST2 dataset. The first row shows an incorrect prediction, the second row shows the correct one. Red is negative, blue is positive.

In the first example, we show three visualizations from the MUTAG dataset in Figure 2. The first row represents a correctly predicted instance. Our model successfully identifies the "NO2" motif as a moderately positive symbol for mutagenicity. The "H" or the carbon ring is considered a negative sign for mutagenicity. Once the "NO2" and the ring join, they become a strong positive symbol for mutagenicity. This phenomenon is consistent with the knowledge that the carbon rings and "NO2" groups tend to be mutagenic (Debnath et al., 1991). We check instances with wrong predictions and show two representative examples. From the second row in Fig. 2, the GCN model precisely finds out the "NH2" motif with the ring motif as a strong mutagenic symbol. But another wandering part without connection shows a strong non-mutagenic effect, ultimately leading to an incorrect prediction. The second row shows another typical failure pattern. The model catches the "NH2" and part of the carbon ring as a mutagenic symbol, but the "CH3" on the bottom right shows a strong non-mutagenic effect. The model erroneously learns a negative interaction between them.

In the second example, we show two visualizations for the Graph-SST2 dataset in Figure 3. The sentence in the first row is labeled negative, yet its prediction is wrong. Our algorithm can explain the decision that the GNN model regards first half of the sentence ("Maybe it's asking too much") as negative, the second half ("going to inspire me", "want a little more than this") as positive. But the model can not tell the subjunctive tone behind the word "if", and consequently yields a positive but incorrect prediction. The sentence in the second row is negative, and the prediction is correct. Our algorithm precisely identifies the positive part ("Though Ford and Neeson capably hold our interest") and the negative part ("but its just not a thrilling movie"). Moreover, it reveals that the GCN model can correctly learn the transition relationship between these two components.

We observe that our method can detect non-linear interactions between subgraphs throughout the agglomeration process from above examples. It can help to diagnose the incorrect predictions and enhance the credibility of the model. More visualizations and efficiency study are in Appendix E.

## 6    CONCLUSIONS

In this work, we present DEGREE which explains a GNN by decomposing its feedforward propagation process. After summarizing the fundamental rules for designing decomposition based explanation, we propose concrete decomposition schemes for those commonly used layers in GNNs. We also design an algorithm to provide subgraph-level explanation via agglomeration, which efficiently employs the topological information in graphs. Experimental results show that DEGREE outperforms baselines in terms of faithfulness and can capture meaningful structures in graph data.

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

# A  DATASETS AND EXPERIMENTAL SETTING

In this section, we introduce the detail of the datasets as wall as the experimental setting. The code can be found at `https://anonymous.4open.science/r/DEGREE-3128`.

**Model setup:** We adopt GCN model and GAT model architecture for corresponding experiments. We use GCN($\#in\_channel, \#out\_channel, activation$) to denote a GCN layer, and similar notation for GAT layer and fully connected layer.

The structure of GCN model for node classification task is following:
GCN($\#$feature$, 20, ReLU$) $-$ GCN($\#$feature$, 20, ReLU$) $-$ GCN($\#$feature$, 20, ReLU$) $-$
FC($20, 20, ReLU$) $-$ FC($20, \#$label$, Softmax$).

For graph classification task, we adopt a Max-pooling layer before the FC layers:
GCN($\#$feature$, 20, ReLU$) $-$ GCN($\#$feature$, 20, ReLU$) $-$ GCN($\#$feature$, 20, ReLU$) $-$
$Maxpooling -$ FC($20, 20, ReLU$) $-$ FC($20, \#$label$, Softmax$).

For GAT model architecture, we replace all GCN layers with GAT layer and keep the remaining setting unchanged. For experiments on Tree-Grid dataset, we adopt 4-layer GCN/GAT model by adding one more GCN/GAT layer with same setting before FC layers.

**Dataset statistic:** The statistics of synthetic datasets and real-world datasets are reported in Table 2.

Table 2: Statistics of the dataset

| Dataset | # of nodes | # of edges | # of graphs | # of labels | features |
|---------|-----------|-----------|-------------|-------------|----------|
| BA-Shapes | 700 | 4,110 | 1 | 4 | Constant |
| BA-Community | 1,400 | 8,920 | 1 | 8 | Generated from Labels |
| Tree-Cycles | 871 | 1,950 | 1 | 2 | Constant |
| Tree-Grid | 1,231 | 3,410 | 1 | 2 | Constant |
| MUTAG | 131,488 | 266,894 | 4,337 | 2 | Node Class |
| Graph-SST2 | 714,325 | 1,288,566 | 70,042 | 2 | BERT Word Embedding |

**Experimental setting:** For all datasets, we use a train/validation/test split of 80%/10%/10%. For all synthetic datasets, the GCN model is trained for 1,000 epochs and the GAT model is trained for 200 epochs. For MUTAG dataset, the GCN and GAT model is trained 30 epochs. For Graph-SST2 dataset, the GCN model is trained 10 epochs. We use Adam optimizer and set the learning rate to 0.005, the other parameters remain at their default values. We also report the accuracy metric reached on each dataset in Table 3.

Table 3: Accuracy performance of GNN models

| Dataset | BA-Shapes | | BA-Community | | Tree-Cycles | | Tree-Grid | | MUTAG | | Graph-SST2 |
|---------|------|------|------|------|------|------|------|------|------|------|------------|
| Task | Node Classification | | | | | | | | Graph Classification | | |
| Model | GCN | GAT | GCN | GAT | GCN | GAT | GCN | GAT | GCN | GAT | GCN |
| Training | 0.96 | 0.98 | 0.99 | 0.83 | 0.91 | 0.93 | 0.85 | 0.83 | 0.80 | 0.81 | 0.91 |
| Validation | 0.97 | 0.96 | 0.88 | 0.85 | 0.90 | 0.92 | 0.84 | 0.84 | 0.78 | 0.80 | 0.90 |
| Testing | 0.93 | 0.94 | 0.87 | 0.83 | 0.89 | 0.92 | 0.81 | 0.80 | 0.77 | 0.79 | 0.88 |

**Hardware setting:** We introduce the hardware that we use for the experiments.

CPU: AMD EPYC 7282 16-Core Processor.

GPU: GeForce RTX 3090 NVIDIA-SMI: 460.32.03 Driver Version: 460.32.03 CUDA Version: 11.2.

# B    ATTENTION DECOMPOSITION

To calculate the attention coefficient, we need to first calculate the pre-normalized attention coefficient between node i and node j as:

$$\tilde{\alpha}_{i,j} = LeakyReLU([X_i[t]W||X_j[t]W]a)$$

And we use $\tilde{\alpha}_{i,j}$ to denote a vector which consist of the pre-normalized attention coefficients between node $i$ and its neighbors. Then we calculate the normalized attention coefficient of node $i$ via $Softmax$ over its neighbors:

$$\alpha_i = Softmax(\tilde{\alpha}_i)$$

We use $Softmax(|\cdot|)$ to measure the dimension-wise magnitude, and let them compete for the original value. The division between two vectors is element-wise.

# C    ALGORITHM

We conclude the computation steps of subgraph-level explanation (Sec 4) in Algorithm 1, 2 and 3.

---

**Algorithm 1:** The algorithm of subgraph agglomeration

---

**Data:** Graph $\mathcal{G} = (\mathcal{V}, \mathcal{E})$, Label $y$, Target model $f$, Hyperparameter $q$.
**Result:** Explanation tree $T$.
**Score Metric Function:** $\phi$ from Algorithm 3.
**Initialization:** score queue $ScoresQ \leftarrow \varnothing$, explanation tree $T \leftarrow \varnothing$.
**for** $v \in \mathcal{V}$ **do**
$\quad | \quad ScoresQ.add\left(\{v\}, priority = \phi(f, y, \{v\})\right)$
**end**
**while** $ScoresQ$ is not empty **do**
$\quad |$   Select Base Subgraph Set $\mathcal{B} = ScoresQ.top(q)$
$\quad |$   T.add($\mathcal{B}$)
$\quad |$   $ScoresQ \leftarrow \varnothing$
$\quad |$   **for** $\mathcal{B}_m \in \mathcal{B}$ **do**
$\quad | \quad |$   Candidate Subgroup Set $\mathcal{C}$ = Get Candidate($\mathcal{G}, \mathcal{B}_m$) with Algorithm 2
$\quad | \quad |$   **for** $c \in \mathcal{C}$ **do**
$\quad | \quad | \quad |$   $ScoresQ.add(c, priority = \phi(f, y, c) - \phi(f, y, \mathcal{B}_m))$
$\quad | \quad |$   **end**
$\quad |$   **end**
$\quad |$   $ScoresQ = |ScoresQ - \mathbb{E}(ScoresQ)|$
**end**
**return** $T$

---

**Algorithm 2:** The algorithm of candidate node set selection

---

**Data:** Graph $\mathcal{G} = (\mathcal{V}, \mathcal{E})$, Subgraph $\mathcal{B}_m$.
**Result:** Candidate Subgraph Set $\mathcal{C}$.
$\mathcal{C} \leftarrow \varnothing$
Neighbour nodes $\mathcal{N} \leftarrow \{\bar{n}|e =<n, \bar{n}>, e \in \mathcal{E}, n \in \mathcal{B}_m, \bar{n} \in \mathcal{V} \setminus \mathcal{B}_m\}$
**for** $v \in \mathcal{N}$ **do**
$\quad |$   $\mathcal{C}.add(\mathcal{B}_m \cup \{v\})$
**end**
**return** $\mathcal{C}$

---

---

**Algorithm 3:** The algorithm of score computation $\phi$

---

**Data:** Graph $\mathcal{G} = (\mathcal{V}, \mathcal{E})$, model $f$, NodeSet $\mathcal{N}$
**Result:** score $\phi(f, y, \mathcal{N})$
Sample a context set $\mathcal{S}$ by Random Walk within the $L$-hop neighbor region of $\mathcal{N}$.
**Return:** $\phi(f^{\mathcal{D}}, y, \mathcal{N}) = \frac{1}{|\mathcal{S}|} \sum_{s \in \mathcal{S}} (f(\mathcal{N} \cup s) - f(s))$

---

## D    EFFICIENCY STUDY

DEGREE is achieved by decomposing the feedforward process of the target model. Thus the efficiency is highly dependent on the model structure. We report the statistic of time consuming on each dataset for GCN and GAT model in Table 4.

We quantified the relationship between the size of the calculation graph and the time taken. The result is reported in Figure 4

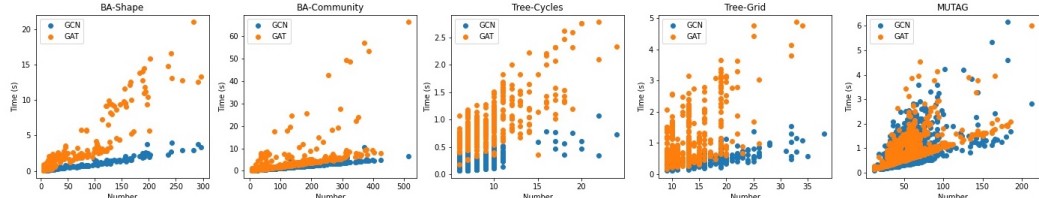

Figure 4: The quantitative studies of efficiency for different datasets and models. For the synthetic datasets, the horizontal coordinate represents the number of nodes in computation graph. For the MUTAG dataset, the horizontal coordinate represents the number of edges in computation graph.

Table 4: Efficiency performance.

| Dataset | BA-Shapes | | BA-Community | | Tree-Cycles | | Tree-Grid | | MUTAG | |
|---|---|---|---|---|---|---|---|---|---|---|
| Model | GCN | GAT | GCN | GAT | GCN | GAT | GCN | GAT | GCN | GAT |
| Avg. Time (s) | 0.44 | 1.98 | 1.02 | 2.44 | 0.25 | 0.96 | 0.37 | 1.03 | 0.83 | 0.79 |

## E    QUALITATIVE EXPERIMENTS EXAMPLES

In this section, we report more qualitative evaluation results on the Graph-SST2 and the MUTAG datasets. The results are reported in Figure 5 and 6.We also investigate the time efficiency of our agglomeration algorithm in terms of the relationship between $q$, the node number and the time spent.

We also investigate the time efficiency of our agglomeration algorithm in terms of the relationship between $q$, the node number and the time spent.

## F    ADDITIONAL EXPERIMENTS FOR REBUTTAL

### F.1    QUANTITATIVE EVALUATION

In this section, we perform additional experiments comparing DEGREE with GNN-LRP Schnake et al. (2020) and SubgraphX Yuan et al. (2021). We use the MUTAG dataset and the Graph-SST2 dataset, as presented in Sec 5.1. The target model is the same as that introduced in Sec 5.2.2. Note that we modify our method to search only for nodes that boost the score of the class of interest. We employ the ACC Liang et al. (2020) as an evaluation metric. The ACC reflects the consistency of predictions based on the whole graph and between interpreted subgraphs. Thus, ACC does not

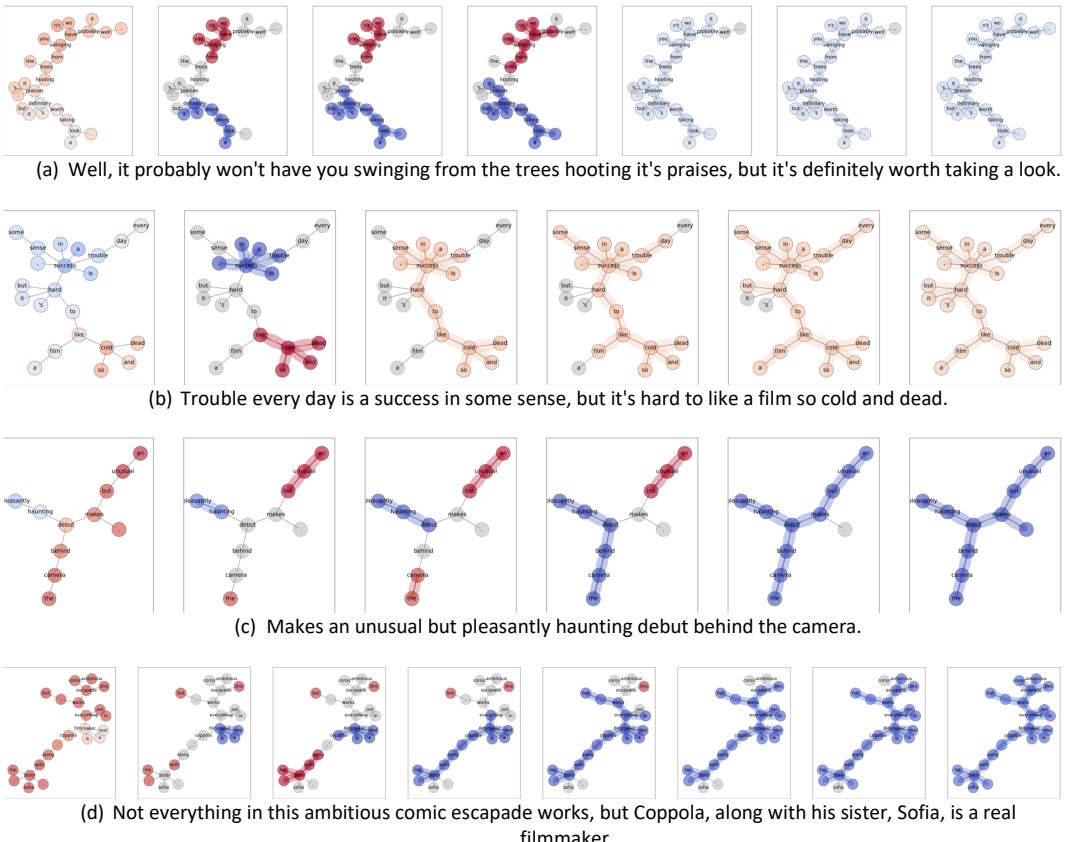

(a) Well, it probably won't have you swinging from the trees hooting it's praises, but it's definitely worth taking a look.

(b) Trouble every day is a success in some sense, but it's hard to like a film so cold and dead.

(c) Makes an unusual but pleasantly haunting debut behind the camera.

(d) Not everything in this ambitious comic escapade works, but Coppola, along with his sister, Sofia, is a real filmmaker

Figure 5: The subgraph agglomeration results on the Graph-SST2 dataset with a GCN graph classifier. All instances all correctly predicted. Red is negative and blue is positive.

require ground truth label. We further define the sparsity as the ratio of the size of the explanation subgraph to the original graph. At the same sparsity, the higher the ACC, the better the interpretation. Figure 8 shows the ACC of DEGREE, GNN-LRP and SubgraphX under various sparsity. We can find that DEGREE has competitive performance compared to GNN-LRP and SubgraphX. Besides, DEGREE has better time efficiency.

## F.2 QUALITATIVE COMPARISON

In this section we make a qualitative comparison between DEGREE and SubgraphX. We randomly select a number of similar molecules and visualize the explanations generated by DEGREE and SubgraphX. We report them in the Figure 9. We can find that none of the subgraphs generated by SubgraphX include the 'N-H' or 'N-O'. They only select the carbon ring as the important part. In contrast, DEGREE can precisely indicate that the mutagenicity is caused by the 'N-H' or 'N-O'.

## F.3 FORWARD-LOOKING EXPERIMENT

In this section, we present a simple prospective experiment from the early stages of this work. The dataset was generated by modifying the MUTAG dataset by selecting half of the graphs in the dataset and picking a node at random in each graph, giving it a special feature value of 1 while giving the other nodes a background white noise feature. Our task is to predict whether a graph contains special nodes or not. We train a 3-layer GCN which achieves 100% accuracy. We then use DEGREE to calculate the contribution score for each node. DEGREE is able to locate special nodes with 100% accuracy. Figure 10 shows the visualisation.

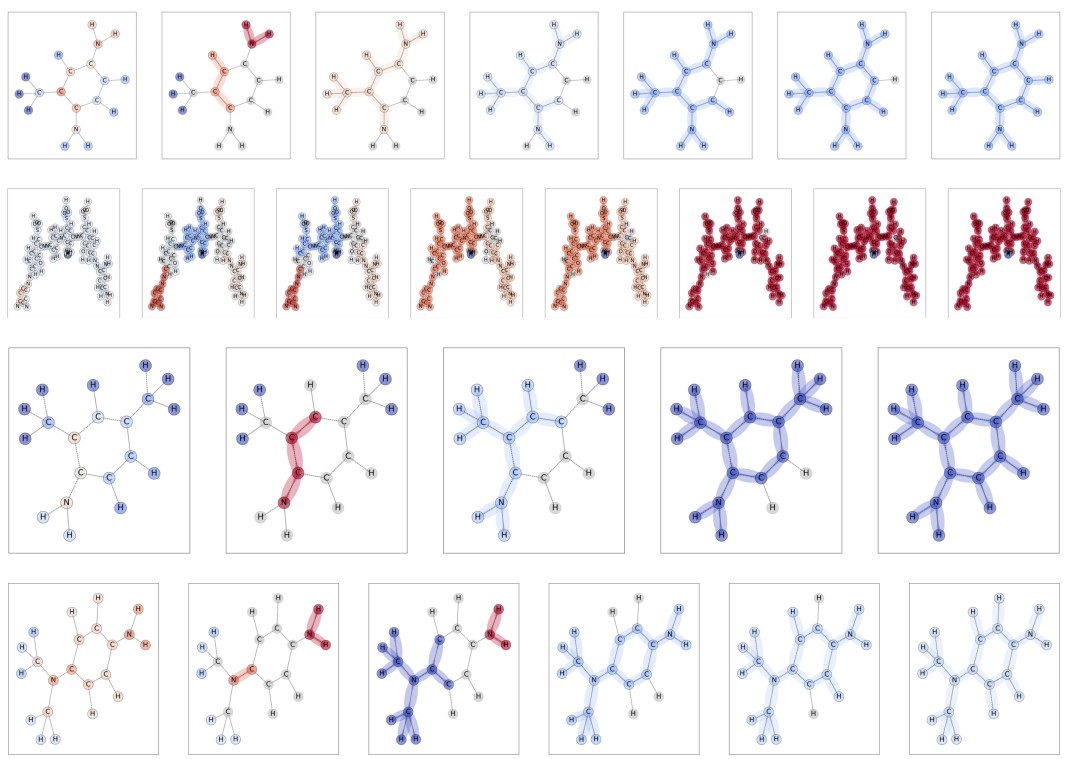

Figure 6: The subgraph agglomeration results on MUTAG dataset with GCN graph classifier. All instances are incorrectly predicted. Red is mutagenic, blue is non-mutagenic, gray is not selected. The colored edges link the selected nodes.

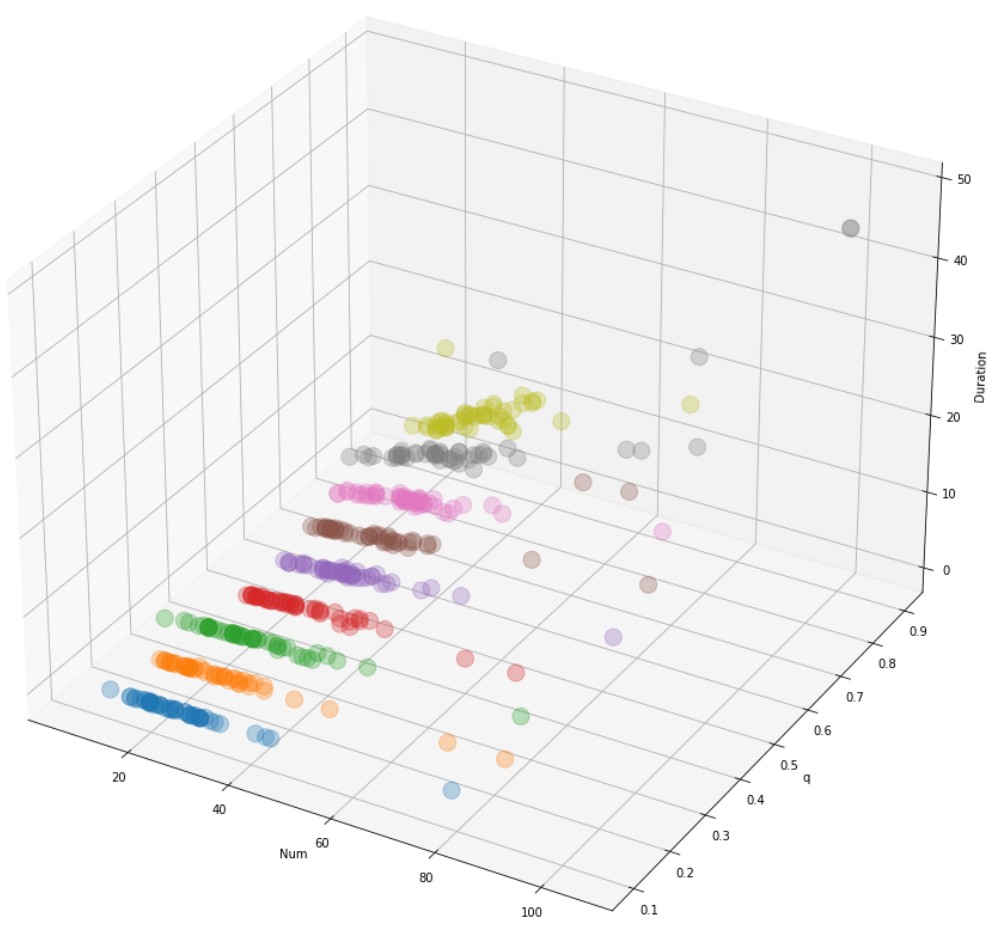

Figure 7: Relationship between the time efficiency(s), graph size and q on the MUTAG dataset.

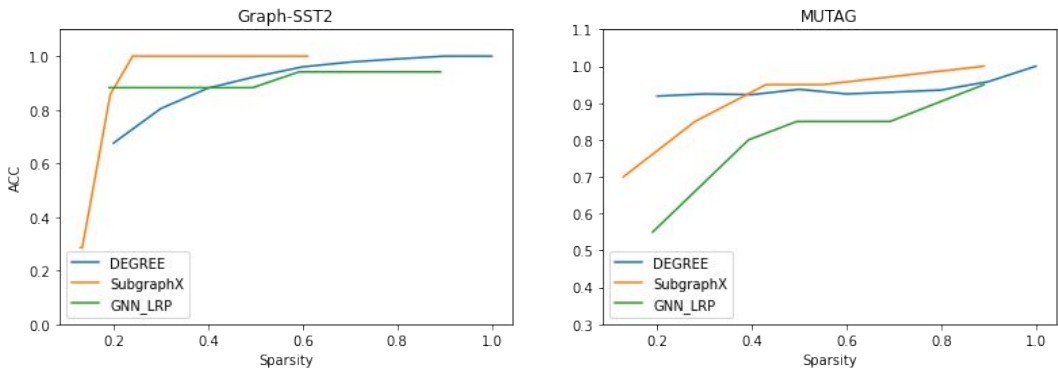

Figure 8: ACC of DEGREE, GNN-LRP and SubgraphX on MUTAG and Graph-SST2.

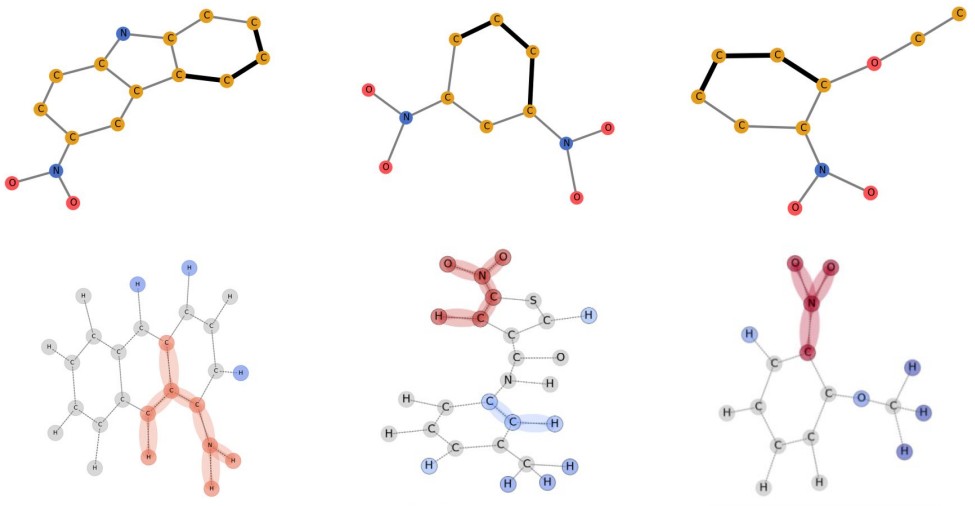

Figure 9: Qualitative comparison of DEGREE and SubgraphX. The first row shows the interpretation generated by SubgraphX. The second row is generated by DEGREE. The red color indicates mutagenicity.

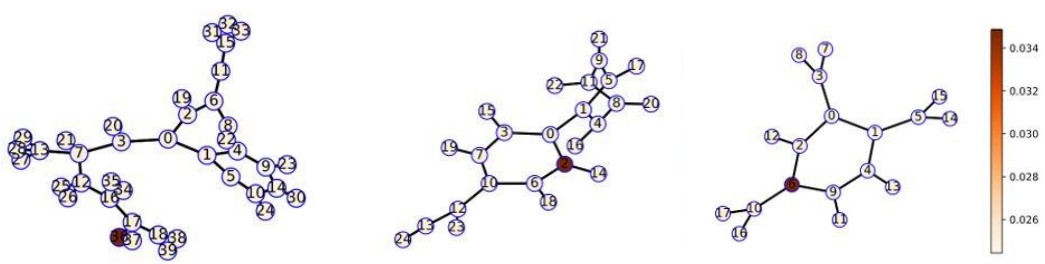

Figure 10: Visualization of the forward-looking experiment. DEGREE can locate the special node with 100% accuracy.

