# OpenReview forum: "DEGREE: Decomposition Based Explanation for Graph Neural Networks"
_ICLR.cc/2022/Conference — ICLR 2022 Poster_

### Official Review · Reviewer_XaeP · 2021-10-26

**Correctness:** 3
**Technical Novelty And Significance:** 2
**Empirical Novelty And Significance:** 3
**Recommendation:** 6
**Confidence:** 3

**Main Review:**

Strengths:
- The problem studied is of importance and interesting.
- The paper is generally well organized.
- Quantitative experiments are done, which is a plus for subjective topics like XAI.

Weaknesses:
- Concepts are not defined or hard to follow. See details.
- Qualitative experiments are hard to follow. See details.
- Questions/unclear points regarding technical details. See details.
- The decomposition method seems straightforward, especially the GCN decomposition (as it is largely linear).

Details.
- Some technical concepts are not defined or hard to follow.
    - Section 3.1. The goal of explanation is to find the most important subgraph in $\mathcal{G}$ given $f(\mathcal{G})$. The concept is vaguely defined. What is the concept of 'important'? Further, in the experiments, the task is actually to predict whether a node/edge is important or not. This seems to deviate from the definition.
    - Section 4. 'We could compute the contribution score of any given node groups'. However, the notion of 'contribution score' is not defined previously. This makes this section hard to understand.
- Qualitative experiments are hard to follow.
    - Figure 2. What does it mean 'the process goes from left to right'?
    - The authors claim that 'existing works suffer from adversarial triggering issues (perturbation) or inaccuracy (approximation)'. However, the claim is not supported, especially the perturbation-based drawback of unrealistic structures. I think this point can be better supported by showing examples that perturbation-based methods indeed generate such samples. At this point, I find it hard to evaluate the qualitative experiments.
- Questions regarding technical details.
    - Section 3.1, $f: \mathcal{G}\mapsto \mathbb{R}^{|\mathcal{V}|}$ or $\mathbb{R}$. It seems that the formulation only works on binary classification or regression. Can this framework (respectively, DEGREE) be extended to the multi-class setting?
    - Section 3.3 Intuition (2): ideally there should be little interaction between the target portion and the background portion. However, in graphs, this intuition can hardly be satisfied because nodes are connected to each other. How does the framework work when this intuition does not hold? Or can you provide evidence that this intuition somehow holds in real-world data?
    - Equation 13, this equation is hard to understand. First, what does it mean $softmax(\mathbf{X}[t])$? Second, what does it mean $\exp(|\mathbf{X}^\gamma[t]|)$? Third, what does it mean to divide vectors with vectors (given that $\mathbf{X}[t]$ is a vector)?

- Minor issues.
    - Please change the coloring of the figures such that they are visible when printed out.
    - Section 4.2, how should we initialize $\mathcal{E}$, $\mathcal{S}_i$? How should we tune the parameters $I, q$?
    - How is the definition of 'importance' in Graph-SST2?
    - Section 5, last paragraph, 'detect non-linear interactions', what does it mean non-linear interactions? Does it mean language conjunctions such as 'if', 'but'?

**Summary Of The Paper:**

This paper proposes a decomposition-based explanation method for graph neural networks. The motivation of this paper is that existing works based on approximation and perturbation suffer from drawbacks. To address the issue of existing works, the authors directly decompose the influence of node groups in the forward pass. The decomposition rules are designed for GCN and GAT. Further, to efficiently select subgraph groups from all possible combinations, the authors propose a greedy approach to search for maximally influential node sets. Experiments on synthetic and real-world datasets verify the improvements over existing works.

**Summary Of The Review:**

In summary, this paper has its merits. The paper tackles an interesting problem. The proposed framework is generally sound. The paper is organized well in general. However, given its unclear points in experiments and technical details, I do not recommend acceptance at this point.

====Post Response====

My doubts and suggested weaknesses are addressed partly. I am willing to suggest borderline accept.

---

> ### Author Response · Authors · 2021-11-20
> **Response to Reviewer XaeP (Part 1/2)**
>
> We thank the reviewer for the thoughtful suggestions and detailed reviews. We also appreciate your recognition of the significance of our contribution. Here are our response to your concerns.
>
> ### Q1. Some technical concepts are not defined or hard to follow.
> #### __Q1.1 What is the concept of 'important'? The notion of 'contribution score' was not defined previously.__
>
> The 'contribution score' is defined as the target portion of the logit of the particular class we are interested in. It can be calculated in accordance with section 3. The larger the target portion contributed by a node group, the more important it is.
>
> #### __Q1.2 Further, in the experiments, the task is actually to predict whether a node/edge is important or not. This seems to deviate from the definition.__
>
> DEGREE has the ability to calculate the contribution score of any group of nodes. Thus, the contribution score of a node/edge can be defined as the contribution score of a special node group containing only one/two nodes. We define the contribution score of an edge as the contribution score of the group of nodes containing its two vertices.
> To compare with the baselines methods, we use DEGREE to calculate the contribution score of node/edge and adopt AUC as metric.
>
> DEGREE can further be extended to search for important subgraph via the agglomeration method introduced in Sec 4. We additionally conduct a subgraph level quantitative experiments. We chose one of the decomposition based methods, GNN-LRP and the subgraph-level interpretation search algorithm, SubgraphX [1], to be compared with DEGREE on MUTAG dataset and Graph-SST2 dataset.  Note that we modify our method to only search nodes that boost the score. We employ ACC [2] as the evaluation metric. The ACC is calculated as below:
> - 1.The explanation method outputs the important nodes/walk/subgraph to be identified.
> - 2.Check the consistency between the predictions based on the whole graph and the explanatory subgraphs. If the prediction based on the whole graph and the explanatory subgraphs is the same, then the consistency score is 1.
> - 3.Calculate ACC as the average consistency score over testing samples.
>
> We depict the relationship between explanation results’ ACC and Sparsity. The Sparsity is the ratio of the size of the explanation subgraph to the original graph. The higher the ACC, the better the interpretation. We can see that DEGREE and SubgraphX have comparable performance. Here are the Anonymous link to the comparison figures.
>
> Comparison on MUTAG dataset. [ACC_MUTAG](https://anonymous.4open.science/r/ICLR_Rebuttal_FIG-1942/ACC_MUTAG.png).\
> Comparison on Graph-SST2 dataset. [ACC_SST](https://anonymous.4open.science/r/ICLR_Rebuttal_FIG-1942/ACC_SST.png).
>
> [1] Yuan, Hao, et al. "On explainability of graph neural networks via subgraph explorations." arXiv preprint arXiv:2102.05152 (2021).\
> [2] Liang, Jian, et al. "Adversarial infidelity learning for model interpretation." Proceedings of the 26th ACM SIGKDD International Conference on Knowledge Discovery & Data Mining. 2020.
>
> ### Q2. Qualitative experiments are hard to follow.
> #### __Q2.1 What does it mean 'the process goes from left to right'?__
> Figure 2 shows the process of the agglomeration (introduced in Sec 4) step by step from left to right. The leftmost one shows the contribution score of individual nodes, while the rightmost one shows the contribution score of the whole graph.
> #### __Q2.2 The perturbation-based drawback of unrealistic structures.__
> Perturbation-based methods can introduce artefacts (unrealistic structures) that lead to unreliable interpretations [1] as the perturbed data becomes inconsistent with the original dataset [2]. Take this figure from the MUTAG dataset as an example.
>
> The Anonymous link to the MUTAG example. (https://anonymous.4open.science/r/ICLR_Rebuttal_FIG-1942/3369.png)
>
> The original prediction is mutagenic. If we remove node 0 and node 2 from the graph and get a perturbed graph (artefact) that could not exist in nature, the prediction becomes non-mutagenic. However, the node 0 and node 2 are not in N-O or N-H (the ground truth we have). This means that the artefacts introduced affect the quality of the interpretation.
>
> [1] Fong, Ruth C., and Andrea Vedaldi. "Interpretable explanations of black boxes by meaningful perturbation." Proceedings of the IEEE international conference on computer vision. 2017.\
> [2] Qiu, Luyu, et al. "Resisting out-of-distribution data problem in perturbation of xai." arXiv preprint arXiv:2107.14000 (2021).

---

> ### Author Response · Authors · 2021-11-20
> **Response to Reviewer XaeP (Part 2/2)**
>
> ### Q3. Questions regarding technical details.
> #### __Q3.1 Can this framework (respectively, DEGREE) be extended to the multi-class setting?__
> Yes, the contribution score is defined as the target portion of the logits of the class of interest. Thus it can be extended on the multi-class setting.
> #### __Q3.2 How can intuition hold because nodes are connected to each other?__
> GNNs use an aggregation mechanism to aggregate information between nodes. It is essentially a summation, and different GNNs have different weighting methods and objects of the summation. For example, GAT uses attention coefficient as weighting.  The summation is separable, thus the information flow in aggregation could be separable. The intuition that information aggregation is distributive in nature is also demonstrated by Eq. 4 in the Sec 3.3.
>
> In the early stages of this work, we conducted a simple prospective experiment. We modified the MUTAG dataset by selecting half of the graphs in the dataset and randomly picking a node in these graphs and giving it a special feature value while giving the other nodes a background feature. A 3-layer GCN was trained to predict whether a graph contained a special node, and then DEGREE was used to find the node with special value; DEGREE was able to locate the special node with 100% accuracy. This simple experiment illustrates DEGREE's ability to distinguish the target portion of the information from the background portion. There is still some information that may interact with each other, and we will explore how to decouple them in future work.
>
> #### __Q3.3 Equation 13, this equation is hard to understand.__
> We thank the reviewer for pointing out the unclearness. The notation in the paper is not appropriate. As shown in Eq. 11, to calculate the attention coefficient, we need to first calculate the pre-normalized attention coefficient between node i and node j as:
>
> $$
> \tilde{\alpha}_{i,j} = LeakyReLU([X_i[t]W || X_j[t]W]a)
> $$
>
> And we use $ \tilde{\alpha}_{i,j}$
> to denote a vector which consist of the pre-normalized attention coefficients between node $i$ and its neighbors. Then we calculate the normalized attention coefficient of node $i$ via $Softmax$ over its neighbors:
>
> $$
> {\alpha}_{i} = Softmax(\tilde{\alpha}_i)
> $$
>
>  We use $Softmax(|\cdot|)$ to measure the dimension-wise magnitude, and let them compete for the original value. The division between two vectors is element-wise.
>
> ### Q4.Minor issues.
> #### __Q4.1 Please change the coloring of the figures such that they are visible when printed out.__
> We will update the coloring and add color bar to make it easy to read.
> #### __Q4.2 How should we initialize $\mathcal{E}$ and $\mathcal{S}_{i}$. How should we tune the parameters $I$, $q$.__
> At the beginning of our algorithm, the $\mathcal{B}_{m}$ is an empty set, and starts by measuring the contextual contribution of each individual node. We select nodes with large score  magnitude to form $\mathcal{S}_1$.
>
> We use $q$ to control the ‘granularity’ of the agglomeration process, i.e. the larger q will let the agglomeration choose less nodes to be merged. The process will terminate until all nodes are included, and $I$ is the number of $\mathcal{S}$. While we can also set a maximum $I$ as the budget and stop the search early.
>
> #### __Q4.3 How is the definition of 'importance' in Graph-SST2?__
> There is no ground truth of important words in Graph-SST2. We conducted another quantitative experiment without ground truth labels, which is shown in Q1.2.
>
> #### __Q4.4 What does it mean non-linear interactions?__
>  Yes, the non-linear interactions mean language conjunctions such as ‘if’, ‘but’.
>
> We thank the reviewers for the recognition of our merits. We hope that our response has addressed the points that are unclear. Thank you!

---

> ### Author Response · Authors · 2021-11-22
> **Looking forward to your feedback.**
>
> Dear Reviewer XaeP ,
>
> Thanks again for your valuable suggestions! We have responded to your initial comments. We are looking forward to your feedback and will be happy to answer any further questions you may have.
>
> Thank you, author

---

> ### Comment · Reviewer_XaeP · 2021-11-29
> **Thanks for author responses.**
>
> The author responses clear up some of my questions. I am willing to revise my rating to 6, i.e. borderline accept.

---

### Official Review · Reviewer_eChJ · 2021-11-02

**Correctness:** 4
**Technical Novelty And Significance:** 3
**Empirical Novelty And Significance:** 3
**Recommendation:** 8
**Confidence:** 4

**Main Review:**

The paper focuses on the explanation of GNN performance, which is an interesting and critical problem. The paper proposes a reasonable method for the problem to decomposite the graph. The experiments are also comprehensive. However, there are still some points required to be clarified.

1,  The new scoring function proposed in the paper uses a random walk process, so will the calculation process has a high time complexity and space complexity (storing sample graphs)?

2, In q · maxv′ r(v′), what is the impact of the hyper-parameter q on the subgraph expansion?

3, The AUC used in the experiments should be explained in detail. For me, I have to search previous works to understand how to use it in GNN explanation issue. It is not friendly to common readers.



**Summary Of The Paper:**

The paper provides DEGREE, which decomposes the feedforward propagation mechanism of a GNN to understand it. They give realistic decomposition techniques for those typically used layers in GNNs after presenting the key guidelines for developing decomposition-based explanations. They also devise an approach for providing subgraph-level explanation via agglomeration, which makes effective use of graph topology. DEGREE surpasses baselines in terms of fidelity and can capture important structures in graph data, according to experimental results.

**Summary Of The Review:**

The paper is easy to follow and has a reasonable method. Besides, the comprehensive experiments, especially the visualization of results, verify the effectiveness of the proposed method. I surely recommend acceptance after my above doubt is solved.

---

> ### Author Response · Authors · 2021-11-20
> **Response to Reviewer eChJ**
>
> We thank the reviewer for the thoughtful suggestions and detailed reviews. We also appreciate your recognition of the significance of our contribution. Here are our response to your concerns.
>
> ### Q1. Will the calculation process have a high time complexity and space complexity (storing sample graphs)?
>
> We control the sampling process by adjusting its sampling number and the maximum depth. Note that we set the maximum depth to the number of layers of the GNN, as the GNN only aggregates information within the L-hop neighborhood, where L is the number of layers of the GNN. The number of samples taken is within reasonable limits and will not have a high time complexity because the size of  graph in MUTAG and Graph-SST2 is not extremely large. We do not store the intermediate sampling results. We thank the reviewers for pointing out this direction of improvement, as it will help to reduce the sampling time when DEGREE is applied to larger graph datasets than Graph-SST2 and MUTAG.
>
> ### Q2. What is the impact of the hyper-parameter q on the subgraph expansion?
>
> Here q is the threshold for node selection, which will affect the 'granularity' of the agglomeration process, i.e. a larger q will allow agglomeration to select fewer nodes for merging in each step. Thus, q will also affect the agglomeration efficiency. The influence of q is shown in Figure 7 in Appendix.
>
> ### Q3. The AUC used in the experiments should be explained in detail.
>
> We thank the reviewers for pointing out our carelessness. A more detailed name for AUC is AUROC (Area Under Receiver Operating Characteristics). It is a performance measure for classification problems and tells how well the model can distinguish between categories. For consistency with the baseline, we used node-level AUC for the synthetic dataset and edge-level AUC for the MUTAG dataset.
>
> Hope our response can address your concerns. Thank you.

---

> ### Author Response · Authors · 2021-11-22
> **Looking forward to your feedback.**
>
> Dear Reviewer eChJ ,
>
> Thanks again for your valued advice! We have responded to your initial comments. We are looking forward to your feedback and will be happy to answer any further questions you may have.
>
> Thank you, author

---

### Official Review · Reviewer_81sK · 2021-11-02

**Correctness:** 3
**Technical Novelty And Significance:** 3
**Empirical Novelty And Significance:** 3
**Recommendation:** 6
**Confidence:** 3

**Main Review:**

Strengths:
S1. The paper innovatively proposes a new explainable framework that decomposes the information into target and background portion. In contrast of previously used approximation-based, perturbation-based, additive feature attribution methods, the authors claim that the newly proposed method has the advantages of higher fidelity and node-level explainability.
S2. Two popular GNN framework GCN and GAT are used as examples for decomposition operation, which demonstrates the methods compatibility to the current mainstream. A new subgraph construction algorithm is also proposed to tackle the problem arising during finding most important subgraphs, i.e., it’s impossible to enumerate all subgraphs.
S3. The quantitative evaluation shows in two metrics: Explanation AUC and time efficiency. The proposed method significantly outperforms other benchmarked methods in at least one of the metrics (some benchmarked methods are only shown for Explanation AUC, not for time efficiency though), potentially making it a new choice under different use scenario. A qualitative evaluation was also included for a visual check of the result quality.
S4. The paper is well organized and clearly written. Figures are provided with descriptive caption, including details needed for comprehension. Datasets includes both synthetic datasets and real-world datasets for diversity.

Weaknesses:
W1. The transition from absolute contribution scores to relative contribution score in Sec 4.1 is abrupt without detailed reasons. Why are relative scores preferred? Also, if they are preferred, why use absolute score in Sec 3? Providing answers to these two questions will make the logic flow more smoothly.
W2. The exclusion of Graph-SST2 dataset from quantitative evaluation raises the question of whether the model has a poor performance on this dataset and hence has little generalization power. Providing the reasons for not including this dataset makes the benchmarking more supportive for the claimed superior model performance.
W3. The limitation of the model was shown in the wrong prediction example in the qualitative evaluation but was not discussed in detail. What could have caused the wrong prediction and how to circumvent them?
W4. Missing related work that should be discussed: On Explainability of Graph Neural Networks via Subgraph Explorations (ICML 2021)

Minor comments:
1. In Figure 2, I assume the shade of color means the value of the score, but it is probably a better idea to explicitly state it in the caption or have a legend in the figure.
2. It would be preferred to include the statistics such as the number of graphs for synthetic datasets as well.




**Summary Of The Paper:**

The paper aims at tackling the black-box nature problem of GNN by introducing a new type of explainable GNN framework called DEGREE (Decomposition based Explanation for GRaph nEural nEtworks). There are mainly two innovations. The first one lies in its ability to track contribution of components in the input graph. The second one is the algorithm for subgraph-level explanation via agglomeration. The model achieves a good compromise between performance and time efficiency.

**Summary Of The Review:**

The paper innovatively proposes to decompose the information flow for explainability and provides a new algorithm to construct subgraphs to reveal more complex interactions. This novel way of explaining graph component contributions give insights into a new explainable GNN framework and is potentially inspiring for other works. The experiments also show its superior performance and time efficiency compared with other methods (although some related work is missing). Thus, I am slightly positive of this paper.

====Post Response====

After reading the author response, they have indeed addressed some of my concerns. However, after also reading through the other reviews and their respective replies, I am inclined to keep the score of "marginally above the acceptance threshold" because I am still more positive towards the work. I have additionally increased the novelty and significance from a 2 to a 3. Thank you for providing the reply, especially the details regarding my original concern about Graph-SST2.

---

> ### Author Response · Authors · 2021-11-20
> **Response to Reviewer 81sK**
>
> We thank the reviewer for the thoughtful suggestions and detailed reviews. We also appreciate your recognition of the significance of our contribution.  Here are our responses to your comments.
> ### Q1. Why are relative scores preferred? Also, if they are preferred, why use absolute score in Sec 3?
>
> The relative score is preferred because a node's score should faithfully reflect its contribution in various situations. It can also be seen as an approximation to the Shapley Value. Take the sentence in the second row of Figure 3 in our paper as an example. The absolute score for the word "movie" is 0.13, while its relative score is -0.04. Since "movie" itself is a very neutral word, the relative score is better (closer to 0).
>
> Thanks to the reviewer for pointing out the unclearness. Sec 3 introduces the principles to decompose GNN layers. We use relative score for all experiments.
>
> ### Q2. The exclusion of Graph-SST2 dataset from quantitative evaluation.
>
> To be consistent with related work, we use the area under the curve (AUC) as a metric. Similar to the accuracy metric in classification tasks, the AUC metric requires ground truth labels. Synthetic and MUTAG datasets naturally have ground truth labels, i.e., which nodes are important. For the synthetic dataset, we consider the nodes within the pattern to be important. For the MUTAG dataset, the "N-H" and "N-O" edges are important. However, Graph-SST2 is a sentiment graph dataset, and each graph is transformed from sentences using the Biaffine parser. Unfortunately, there are no ground truth labels about which nodes (words) are important. Therefore, we did not include the Graph-SST2 dataset in our quantitative experiments.
>
> Therefore, we conducted another quantitative experiment without ground truth labels. We chose one of the decomposition based methods, GNN-LRP and the subgraph-level interpretation search algorithm, SubgraphX, to be compared with DEGREE on MUTAG dataset and Graph-SST2 dataset.  Note that we modify our method to only search nodes that boost the score. We employ ACC [1] as the evaluation metric. The ACC is calculated as below:
> - 1.The explanation method outputs the important nodes/walk/subgraph to be identified.
> - 2.Check the consistency between the predictions based on the whole graph and the explanatory subgraphs. If the prediction based on the whole graph and the explanatory subgraphs is the same, then the consistency score is 1.
> - 3.Calculate ACC as the average consistency score over testing samples.
>
> We depict the relationship between explanation results’ ACC and Sparsity. The Sparsity is the ratio of the size of the explanation subgraph to the original graph. The higher the ACC, the better the interpretation. We can see that DEGREE and SubgraphX have comparable performance. Here are the Anonymous link to the comparison figures.
>
> Comparison on MUTAG dataset. [ACC_MUTAG](https://anonymous.4open.science/r/ICLR_Rebuttal_FIG-1942/ACC_MUTAG.png).\
> Comparison on Graph-SST2 dataset. [ACC_SST](https://anonymous.4open.science/r/ICLR_Rebuttal_FIG-1942/ACC_SST.png).
>
> [1] Liang, Jian, et al. "Adversarial infidelity learning for model interpretation." Proceedings of the 26th ACM SIGKDD International Conference on Knowledge Discovery & Data Mining. 2020.
>
> ### Q3. What could have caused the wrong prediction and how to circumvent them?
>
> This is a very good question. From our observations of incorrect predictions, we have empirically concluded that there are two reasons for the errors. The first possible reason is shown by the second row in Figure 2. The overhanging sulphuric acid (the blue cross-shaped ion in the middle) shows a strong non-mutagenic effect, which ultimately leads to the wrong prediction. A second possible cause is evidenced by the third row in Figure 2. Most of the non-mutagenic samples contained methyl (CH3), so the model may have incorrectly derived strong evidence for non-mutagenicity. In our appendix, the second row of Fig. 6 shows the incorrect prediction due to cause 1, while the first, third and fourth rows of Figure 6 show the incorrect prediction due to cause 2.
>
> How to circumvent them is also a good question. The output interpretation (contribution score) is derivable thanks to the feed-forward mechanism of DEGREE. This means that we can calibrate the behavior of the model by adding appropriate regularization terms to inject reasonable (ground truth knowledge). This is an interesting and worthwhile direction to explore in the future.
>
> ### Q4. Missing work that should be discussed: On Explainability of Graph Neural Networks via Subgraph Explorations (ICML 2021)
>
> We add comparison quantitative experiment in Q2.
>
> ### Q5. Minor comments
>
> We will update the color bar and legend as the reviewer suggests. The statistics of the datasets are summarized in Table 2 in the Appendix due to the page limitation. There is only one graph in each synthetic dataset.

---

> ### Author Response · Authors · 2021-11-22
> **Looking forward to your feedback.**
>
> Dear Reviewer 81sK ,
>
> Thanks again for your valued advice! We have responded to your initial comments. We are looking forward to your feedback and will be happy to answer any further questions you may have.
>
> Thank you, author

---

### Official Review · Reviewer_gPGV · 2021-11-05

**Correctness:** 3
**Technical Novelty And Significance:** 2
**Empirical Novelty And Significance:** 2
**Recommendation:** 6
**Confidence:** 3

**Main Review:**


The main concern of this paper is the decomposition assumption.  Because of the nonlinearity property in neural networks, how to decompose to the target portion and the background portion for the input features/representations.  It likely has potential interactions between the target portion and the background portion.

As for the faithfulness issue, there exist a few decomposition-based explanations methods, e.g.,  Layer-wise Relevance Propagation (LRP) [1], Excitation BP [2], and GNN-LRP [3]. Most of their methods are proposed to employ score decomposition.

The subgraph-level interpretation search algorithm has been already explored by others [4]. It would be better if authors can compare with them.

[1] Explainability techniques for graph convolutional networks

[2] Explainability methods for graph convolutional neural networks

[3] Higher-order explanations of graph neural networks via relevant walks

[4] On Explainability of Graph Neural Networks via Subgraph Explorations


It is not easy to understand the description in the qualitative evaluation section, especially the description of Figures 2 and 3.
Meanwhile, it would be better if authors can compare with other explanations methods in the qualitative evaluation section.


**Summary Of The Paper:**

This paper proposes a decomposition-based explanations method for graph neural networks.  In detail, the authors design a subgraph level interpretation algorithm to reveal complex interactions between graph nodes, so as to achieve the faithful explanation for GNN predictions. They demonstrate the effectiveness of the proposed method on synthetic and real-world datasets.



**Summary Of The Review:**

The main concern of this paper is the decomposition assumption.  Because of the nonlinearity property in neural networks, how to decompose to the target portion and the background portion for the input features/representations.  It likely has potential interactions between the target portion and the background portion.

---

> ### Author Response · Authors · 2021-11-20
> **Response to Reviewer gPGV**
>
> We thank the reviewer for the detailed reviews. Here are our responses to your comments. \
> ### Q1. How to decompose to the target portion and the background portion for the input features/representations.
> We agree with the reviewer that it is nontrivial to decompose the target portion and the background portion due to the nonlinearity. As shown in Eq. 10, we adopt the telescoping sum as the linearization technique [1]. It can not completely split the target portion and background portion as it can be viewed as an approximation to Shapley value. The experimental results demonstrated the rationality of our approach.
> Our interpretation process is bottom-up, i.e., the decomposed parts (relevant and irrelevant) in a lower layer flow into the higher layer, and can still be kept as decomposed. If we could distinguish between target and background parts in a lower layer, then we could do the same in a higher layer, towards the output layer.
> We also conducted a simple forward-looking experiment in the early stages of this work. We modified the MUTAG dataset by selecting half of the graphs in the dataset and picking a random node in each graph and giving it a special feature value, while giving the other nodes a background feature. A 3-layer GCN is trained to predict whether a graph contains special nodes, and DEGREE is used to find special nodes; DEGREE can locate special nodes with 100% accuracy. This simple experiment can show the ability of DEGREE to distinguish the target portion information from the background portion.
>
> [1] Murdoch, W. James, Peter J. Liu, and Bin Yu. "Beyond word importance: Contextual decomposition to extract interactions from LSTMs." arXiv preprint arXiv:1801.05453 (2018).
>
> ### Q2. Compare with GNN-LRP and Subgraphx [2].
> We thank the reviewers for pointing out the existence of a number of decomposition-based interpretation methods. DEGREE differs from these methods in two main ways. Firstly, DEGREE is a forward decomposition method, whereas LRP, Excitation BP and GNN-LRP are backward decomposition methods. Secondly, DEGREE can reveal the importance of a set of nodes, whereas the backward decomposition method can only show the importance of individual nodes. It is worth noting that by using DEGREE, the importance of a group of nodes is not the sum of the importance of the nodes in that group. Therefore, DEGREE also has the ability to explore sub-graph level interpretations.
>
> We chose one of the decomposition based methods, GNN-LRP and the subgraph-level interpretation search algorithm, SubgraphX, to be compared with DEGREE on MUTAG dataset and Graph-SST2 dataset.  Note that we modify our method to only search nodes that boost the score. We employ ACC [3] as the evaluation metric. The ACC is calculated as below:
> - 1.The explanation method outputs the important nodes/walk/subgraph to be identified.
> - 2.Check the consistency between the predictions based on the whole graph and the explanatory subgraphs. If the prediction based on the whole graph and the explanatory subgraphs is the same, then the consistency score is 1.
> - 3.Calculate ACC as the average consistency score over testing samples.
>
> We depict the relationship between explanation results’ ACC and Sparsity. The Sparsity is the ratio of the size of the explanation subgraph to the original graph. The higher the ACC, the better the interpretation. We can see that DEGREE and SubgraphX have comparable performance. Here are the Anonymous link to the comparison figures.
>
> Comparison on MUTAG dataset. [ACC_MUTAG](https://anonymous.4open.science/r/ICLR_Rebuttal_FIG-1942/ACC_MUTAG.png).\
> Comparison on Graph-SST2 dataset. [ACC_SST](https://anonymous.4open.science/r/ICLR_Rebuttal_FIG-1942/ACC_SST.png).
>
> [2] Yuan, Hao, et al. "On explainability of graph neural networks via subgraph explorations." arXiv preprint arXiv:2102.05152 (2021).\
> [3] Liang, Jian, et al. "Adversarial infidelity learning for model interpretation." Proceedings of the 26th ACM SIGKDD International Conference on Knowledge Discovery & Data Mining. 2020.
> ### Q3.  Qualitative evaluation comparison with SubgraphX.
> We report some qualitative results of SubgraphX. The important edges are in bold. And the important edges selected by SubgraphX fail to include the N-H / N-O (the ground truth label).
>
> Two SubgraphX explanation examples on MUTAG dataset. [SubgraphX_10](https://anonymous.4open.science/r/ICLR_Rebuttal_FIG-1942/subgraphx_10.png).
> [SubgraphX_78](https://anonymous.4open.science/r/ICLR_Rebuttal_FIG-1942/subgraphx_78.png).
>
> Whereas DEGREE can locate them as shown in Figure 2 in our paper.
>
> Hope this will address your concerns.

---

> ### Author Response · Authors · 2021-11-22
> **Looking forward to your feedback.**
>
> Dear Reviewer gPGV ,
>
> Thanks again for your valued advice! We have responded to your initial comments. We are looking forward to your feedback and will be happy to answer any further questions you may have.
>
> Thank you, author

---

> ### Comment · Reviewer_gPGV · 2021-12-07
> **Thanks for author responses**
>
> Most of my concerns have been addressed. I would like to increase my score by one point (from 5 to 6).

---

### Author Response · Authors · 2021-11-22
**General Comments for All Reviewers**


Dear Reviewers,

We are grateful to all reviewers for their many constructive comments and helpful feedback. We are pleased to find that they find our contribution novel (81sK), clearly written (81sK, XaeP), the experiments comprehensive (eChJ, XaeP) and the good visualization(eChJ).

To address your main concerns, we have done our best to improve our work. We have added more baselines (GNN-LRP, SubgraphX) to compare DEGREE quantitatively with their interpretation at the subgraph level, demonstrating that DEGREE is competitive among them. We added a qualitative comparison of DEGREE with SubgraphX to the MUTAG dataset, showing that DEGREE is better at detecting significant subgraphs. We have also updated the color bars in Figures 2 and 3 to improve readability. For details of the experiments, please refer to our detailed answers to these questions or to __Appendix E__ of our revised manuscript. We would like to point out that our proposed method, DEGREE, is non-additive and can therefore discover non-linear relationships between nodes.

We appreciate all the suggestions made by reviewers to improve our work. It is our pleasure to hear your feedback and we look forward to answering your follow-up questions.

Paper4703 Authors

---

### Decision · Program_Chairs · 2022-01-20

**Decision:**

Accept (Poster)

**Comment:**

This paper proposes a decomposition-based explanation method for graph neural networks. The motivation of this paper is that existing works based on approximation and perturbation suffer from various drawbacks. To address the challenges of existing works, the authors directly decompose the influence of node groups in the forward pass. The decomposition rules are designed for GCN and GAT. Further, to efficiently select subgraph groups from all possible combinations, the authors propose a greedy approach to search for maximally influential node sets. Experiments on synthetic and real-world datasets verify the improvements over existing works. During their initial responses, reviewers suggested that the authors experiment with more baselines and also clarify some of the technical details. The authors revised their manuscript to address several of these comments. So, I am tentatively assigning an accept to this paper.